# Isolation and Comprehensive in Silico Characterisation of a New *3-Hydroxy-3-Methylglutaryl-Coenzyme A Reductase 4* (*HMGR4*) Gene Promoter from *Salvia miltiorrhiza*: Comparative Analyses of Plant *HMGR* Promoters

**DOI:** 10.3390/plants11141861

**Published:** 2022-07-16

**Authors:** Małgorzata Majewska, Łukasz Kuźma, Piotr Szymczyk

**Affiliations:** Department of Biology and Pharmaceutical Botany, Medical University of Lodz, Muszyńskiego 1, 90-151 Lodz, Poland; lukasz.kuzma@umed.lodz.pl (Ł.K.); piotr.szymczyk@umed.lodz.pl (P.S.)

**Keywords:** *HMGR4*, microRNA, promoter, *Salvia miltiorrhiza*, transcription factor, transcription factor binding site

## Abstract

*Salvia miltiorrhiza* synthesises tanshinones with multidirectional therapeutic effects. These compounds have a complex biosynthetic pathway, whose first rate limiting enzyme is 3-hydroxy-3-methylglutaryl-coenzyme A reductase (HMGR). In the present study, a new 1646 bp fragment of the *S. miltiorrhiza HMGR4* gene consisting of a promoter, 5′ untranslated region and part of a coding sequence was isolated and characterised in silico using bioinformatics tools. The results indicate the presence of a TATA box, tandem repeat and pyrimidine-rich sequence, and the absence of CpG islands. The sequence was rich in motifs recognised by specific transcription factors sensitive mainly to light, salicylic acid, bacterial infection and auxins; it also demonstrated many binding sites for microRNAs. Moreover, our results suggest that *HMGR4* expression is possibly regulated during flowering, embryogenesis, organogenesis and the circadian rhythm. The obtained data were verified by comparison with microarray co-expression results obtained for *Arabidopsis thaliana*. Alignment of the isolated *HMGR4* sequence with other plant *HMGR*s indicated the presence of many common binding sites for transcription factors, including conserved ones. Our findings provide valuable information for understanding the mechanisms that direct transcription of the *S. miltiorrhiza HMGR4* gene.

## 1. Introduction

*Salvia miltiorrhiza* Bunge, also known as Red sage, or Chinese sage, is an important species used in traditional Chinese medicine. The dried root is used alone or in combination with other herbs to treat various ailments including cardiovascular diseases, menstrual disorders and insomnia [1,2]. In addition, it has been found to have potential in treating cancer [3], Parkinson’s [4] and Alzheimer’s disease [5], as well as renal deficiency [6], hepatocirrhosis [7], acute lung injury [8], fibrosis [9], neuropathic pain [10], diabetes mellitus [11], or alcohol dependence [12]. The main bioactive compounds responsible for such medical properties are quinone diterpenoids (e.g., tanshinones) and phenolic acids. The tanshinones are biosynthesised from intermediates generated in mevalonate (MVA) and methylerythritol phosphate (MEP) pathways [13]. The key rate-limiting enzyme in the MVA pathway converting 3-hydroxy-3-methylglutaryl-coenzyme A (HMG-CoA) to MVA is HMG-CoA reductase (HMGR) [14]. The pivotal role of *HMGR* in plant metabolism is emphasised by the precise regulation of its function at the level of transcription, post-transcription, translation and post-translation [15,16]. To date, five *S. miltiorrhiza HMGR* gene sequences (*HMGR*—*HMGR4*) have been identified and deposited in the GenBank database [17,18,19]. A combination of cDNA sequence similarity searches with exon/intron structure indicates that *HMGR* (EU680958.1) and *HMGR4* (JN831103.1), and *HMGR2* (FJ747636.1) and *HMGR3* (JN831102.1) are probably two pairs of duplicated genes, respectively [19]. Although the coding sequences of the *S. miltiorrhiza HMGR* genes have been identified and described, some of their promoter sequences remain unknown. These include the *HMGR4* promoter.

Promoter sequence analysis provides much valuable information for understanding the regulation of gene expression. Motifs such as the TATA box, CpG islands, tandem repeats or transcription factor binding sites (TFBSs) deserve special attention. Genome-wide analyses indicate that most in vivo functional TFBSs are located in the proximal promoter region [20,21]. These sites form clusters, thus improving interactions of corresponding transcription factors (TFs) to ensure a better execution of their regulatory functions [22]. An essential functional linkage exists between TFs and RNA polymerase II, acting as a large, conformationally flexible multiprotein complex known as a Mediator [23]. This complex regulates polymerase activity by transmitting signals from TFs. Groups of TFs form complex networks of dependencies and act in a coordinated manner in response to intracellular and environmental signals, thus directing many biological processes [24]. Gene expression is also regulated by the activity of microRNAs (miRNAs). They are mainly known as post-transcriptional and translational inhibitors of gene expression. The miRNAs cut the mRNA strands, destabilise the mRNA by shortening its poly(A) tail, and reduce the efficiency of the translation process [25,26]. However, studies on human and *Arabidopsis thaliana* indicate that miRNAs can also regulate gene expression during transcription by binding to promoter sequences [27,28,29,30]. In conclusion, to understand the mechanisms driving gene expression, it is necessary to also understand the nature of the promoter regions. The first step to achieving this goal requires use of bioinformatics tools.

*A. thaliana* is the most widely-studied plant in modern biology. Its wide appeal for scientists results from its fast growth rate, easy maintenance and small space requirements [31]. Moreover, the plant indicates a good tolerance to homozygosity and self-fertility [32]. Genetic studies are attracted by the small size (132 Mbp) of its completely sequenced genome [32]. Due to its similarity to other plants, *A. thaliana* has become the starting point for studying numerous aspects of plant cell-, molecular- and system-biology [33]. The abundance of research conducted on *A. thaliana* has led to the creation of numerous clones, cloning vectors, mutant lines, seeds, databases and online tools containing genomic, epigenomic, transcriptomic and proteomic data [34].

This work describes the isolation of a new *S. miltiorrhiza HMGR4* promoter and 5′ untranslated region (5′UTR) sequences, and their in silico characterisation via specialised databases such as PlantPan 2.0, TSSP and miRBase. Furthermore, the comprehensive in silico analysis presented herein presents valuable new information about the regulatory functions of *HMGR* promoters. The data can be used to create modified or synthetic promoters which could be active under certain controlled conditions [35]. In this way, numerous medically-important metabolites may be obtained.

## 2. Results

### 2.1. In Silico Analysis of the S. miltiorrhiza HMGR4 Promoter

The 1646 bp sequence obtained by the DNA walking method was deposited in GenBank under accession number KT921337.1 (Figure 1). The sequence contained a 51 bp coding region which perfectly coincided with *HMGR4* gene sequence JN831103.1 identified by Ma et al. [19]. The TATA box was located at bases −28 to −33 from TSS. One tandem repeat (at −1296 to −1353) and no CpG islands were found. A pyrimidine-rich sequence (PRS) was recognised in the 5′UTR.

Moreover, 5369 TFBSs and 365 potentially interacting TFs described previously in *A. thaliana* were revealed using the PlantPan 2.0 tool (Appendix A). Each of the obtained TFs could interact with a number of binding sites. The TFBSs were identified at both strands of the examined sequence. The similarity score between the binding sites found in the *S. miltiorrhiza HMGR4* promoter and those detected in *A. thaliana* ranged from 0.7 to 1.0.

Additional analysis of the entire *HMGR4* promoter sequence revealed the following commonly-known consensus sequences: two auxin-responsive elements (AuxREs); seven salicylic acid (SA)-responsive elements, including W box and TCA-elements; two brassinosteroid-responsive elements (BRREs); two ethylene-responsive elements (EREs); two abscisic acid-responsive elements (ABREs); four methyl jasmonate-responsive elements (MJREs); one pathogen-responsive element Eli box 3; three wounding- and pathogen-responsive elements WRE3; three anaerobic-responsive elements (AREs); and twenty-four light-responsive elements (LREs), including ATCT sequences, Box 4, GT1 motif, TCT motifs, GA motifs, AE box, Box I (Figure 1). The vast majority of these results were in agreement with data provided by PlantPan 2.0, which demonstrated the presence of numerous binding sites for TFs responding to these types of stimulation (Appendix A). Only anaerobic-sensitive TFs were not found. The most commonly-observed TFs were those representing the following families: GATA (light); MYB-related and Dof (auxin); WRKY (SA, wounding and pathogen); NAC; NAM (brassinosteroid and abscisic acid (ABA)); MYB-related (ethylene); and CAMTA (MeJa) (Appendix A). The consensus sequences listed above were distributed along the entire *HMGR4* promoter, with some being located only a few nucleotides from each other; this allows more precise regulation of gene expression by dimerization of binding TFs. Such sequences included two SA-responsive TCA elements and two LREs (Table 1).

The search of the PlantPan 2.0 database revealed about 234 TFs that could interact with 915 TFBSs in the *HMGR4* proximal promoter (Appendix A). To complement the findings described above, the binding sites located in the proximal promoter were analysed in terms of their response to external factors. Such data were available for 666 TFBSs (Appendix A). The results indicated that *HMGR*4 transcription may be most dependent on light, SA, bacterial infection and auxins, and less dependent on ABA, gibberellin, chitin, cold or salt stress (Figure 2). The response to these agents was mainly associated with the following TF families: GATA (light); WRKY (SA, bacterial infection, chitin); bZIP (auxins); MYB-related, WRKY and C2H2 (ABA); MYB-related and MADS box (gibberellins); C2H2 and WRKY (cold); and WRKY and MYB-related (salt stress) (Appendix A). To determine the stage of *S. miltiorrhiza* development at which the *HMGR4* gene expression regulation is likely to occur, 450 TFBSs and interacting TFs located in its proximal promoter were examined (Appendix A). The findings indicated flowering, embryogenesis, organogenesis (root, shoot, leaf and flower development) and circadian rhythm (Figure 3).

As *HMGR* genes are crucial for the production of intermediates in the biosynthesis pathway for tanshinones, a literature search was performed for TFs that positively regulate tanshinone production. Following this, based on the results obtained with the PlantPan 2.0 database, the *S. miltiorrhiza HMGR4* promoter sequence was searched for the presence of binding sites for the 20 identified TFs; of these, the results indicated the presence of the following five TFs: BHLH6 (MYC2) (14 binding sites), BHLH74 (2 sites), BZIP20 (32 sites), WRKY2 (8 sites) and WRKY61 (9 sites) (Appendix A) [36,37,38,39,40]. A number of TFBSs were also found to be located in the proximal promoter region. The obtained results suggest that *HMGR4* may play a role in the biosynthesis of tanshinones.

Furthermore, in silico analysis of the *HMGR4* promoter and 5′UTR revealed potentially interacting miRNAs (Table 2). In total, 12 mature miRNAs were found, 8 binding within the promoter and 4 within the 5′UTR.

### 2.2. Microarray and Next-Generation Sequencing (NGS) Co-Expression Data Analysis

The Protein BLAST analysis revealed that the coding sequence of *S. miltiorrhiza HMGR4* (AEZ55673.1) was more similar to *A. thaliana HMGR1* (NP_177775.2), with an identity of 73.76%, than to *A. thaliana HMGR2* (NP_179329.1), with one of 69.48%. Additionally, a phylogenetic study of coding sequences indicated that *HMGR4* from *S. miltiorrhiza* (JN831103.1) and *HMGR1* from *A. thaliana* (NM_106299.4) were more closely related to each other (Appendix A). Therefore, the *A. thaliana HMGR1* gene (At1g76490) was used for further co-expression research. As a result of the conducted microarray analysis, 166 TF genes co-expressed with *A. thaliana HMGR1* in the *r* range of 0.5–1.0 were found: 41 in AtGenExpress Hormone and Chemical Compendium, 37 in AtGenExpress Abiotic Stress Compendium, 34 in AtGenExpress Pathogen Compendium, 25 in AtGenExpress Tissue Compendium, and 29 in AtGenExpress Plus—Extended Tissue Compendium (Appendix A). The RNA-seq analysis did not identify any TF genes co-expressed with *A. thaliana HMGR1* in the WGCNA correlation range of 0.5–1.0.

### 2.3. Comparison of the in Silico HMGR4 Analysis Results with Microarray Co-Expression Data

The comparison identified 32 common TFs in the *S. miltiorrhiza HMGR4* promoter (Table 3), with the most well-represented being TFs from the homeodomain-leucine zipper (HD-ZIP) and WRKY families. The common TFs participated mainly in response to hormones (ABA, ethylene, jasmonic acid and cytokinins), other abiotic factors (light, salt stress, water deprivation, heat and iron ion) and biotic agents (bacteria), embryogenesis, organogenesis (root development), flowering or, finally, tissue development (epidermis) (Table 3).

These 32 common TFs were scanned with the Genomatix Pathway System for the presence of interactions between them. The identified relationships are presented in Figure 4. It was found that SVP, AGL18 and SPL3 proteins were involved together in the regulation of flowering, and PDF2 and ATML1 in epidermal specification in embryos, respectively. Furthermore, NAC3 and RD26 responded to high salinity, drought and ABA, while NAC3 and ZF2 supported resistance to the herbivore *Spodoptera littoralis*. EIL3 and EIL1 played roles in regulating the response to sulphur deficiency and in ethylene signalling, and EBP transcription was light modulated through the EIN2-EIN3/EIL1 pathway.

Of the 32 common TFs that were recognised, 18 were found to interact with the TFBSs situated in the proximal promoter region (Table 3). The ability of these 18 TFs to bind to DNA as dimers or multimers was also tested. The TFBSs identified for HD-ZIP (ATML1, PDF2, HDG1), WRKY (WRKY2, WRKY14, WRKY45, WRKY57, WRKY69) and Dof (DOF5.4) are given in Table 1; all are closely located to each other, and only separated by a few nucleotides. The existence of experimentally-determined interactions between ATML1 and PDF2 proteins was confirmed by the BioGRID database.

### 2.4. Comparison of S. miltiorrhiza HMGR Promoters

The *S. miltiorrhiza HMGR1*, *HMGR2* and *HMGR4* promoter sequences were analysed using the Common TFs tool. Based on the findings, common binding sites for TFs were recognised, and these were classified into 22 matrix families (Figure 5, Table 4), with each single matrix family comprising identical or functionally-similar TFs identified by weight matrices. The following matrix families were found: *Arabidopsis* homeobox proteins (P$AHBP), L1 box (P$L1BX), MYB IIG-type binding sites (P$MIIG), DNA binding with one finger (P$DOFF), GT box elements (P$GTBX), MADS box proteins (P$MADS), MYB-like proteins (P$MYBL), MYB proteins with single DNA binding repeat (P$MYBS), NAC factors with transmembrane motif (P$NTMF), plant specific NAC proteins (P$NACF), transcription repressor KANADI (P$KAN1), W box family (P$WBXF), time-of-day-specific regulatory elements (P$TODS), nodulin consensus sequence 1 (P$NCS1), sweet potato DNA-binding factor with two WRKY domains (P$SPF1), zinc finger proteins (P$ZFAT), light-responsive elements (P$LREM), protein secretory pathway elements (P$PSPE), CGCG box binding proteins (P$CGCG), proteins involved in programmed cell death response (P$PCDR), plant nitrate-responsive elements (P$PNRE) and finally, stomatal carpenter (P$SCAP). As can be seen in Figure 5, the distribution of the matrices within the *HMGR1* and *HMGR2* promoter sequences was strikingly similar. The above TF family analysis found that the *S. miltiorrhiza HMGR1*, *HMGR2* and *HMGR4* genes can be co-regulated in response to abiotic factors (auxins, gibberellins, ABA, SA, jasmonic acid, brassinosteroids, light, water deprivation, salt stress, cold or phosphate starvation), biotic factors (bacteria, fungi and viruses) and during root, stem, leaf and flower organogenesis (Table 4).

The FrameWorker tool indicated the existence of 10,000 10-element-frameworks within the *S. miltiorrhiza HMGR1*, *HMGR2* and *HMGR4* promoters. Two selected models are provided in Figure 6. The frameworks were created based on 52 matrix families common to the tested sequences, some of which are mentioned above in the Common TF results section. The matrix families were located on the positive or negative strand of the promoters.

The DiAlign TF tool analysis found the *HMGR1* (GU367911.1) and *HMGR2* (KF297286.1) promoters to demonstrate the greatest similarity (97%). In contrast, only 17% similarity was found between *HMGR4* (KT921337.1) and *HMGR2*, and 14% between *HMGR4* and *HMGR1*. The greatest number of identical areas was revealed in the proximal fragments of the analysed promoters, as well as in the beginning and the middle of the distal parts. Common overlapping TFBSs were identified in locations where all three tested sequences showed high local similarity; these included two binding sites for *Arabidopsis* homeobox proteins (P$AHBP), one site for SBP domain proteins (P$SBPD), one site for W box family proteins (P$WBXF), one site for DNA binding with one finger factors (P$DOFF), one GT box element (P$GTBX) and, finally, one L1 box (P$L1BX). The PlantPan 2.0 and MatInspector (Genomatix) databases indicated that P$AHBP proteins are mainly involved in the response to hormones (auxins, ABA, cytokinins and gibberellins) and the initiation and development of shoot, root and flower meristems. P$SBPD TFs are associated with inflorescence development, flowering and leaf epidermis differentiation. In turn, P$WBXF matrix family responds to hormonal stimulation (SA, ABA, ethylene and jasmonic acid), other abiotic factors (salt stress, wounding, osmotic stress, heat, water deprivation and cold) and biotic agents (bacteria, fungi and viruses), and also participate in leaf senescence. P$DOFF proteins are primarily involved in the regulation of flowering, circadian rhythm and in response to hormones (auxins and SA). P$GTBX factors participate in the organogenesis of flowers and shoots. In addition, P$L1BX proteins are needed for epidermis development and seed germination. These data are available in Appendix A and Table 4.

### 2.5. The Conservation of Plant HMGR Promoters

MEGA X software alignment of 36 sequences spanning the proximal promoters and 5′UTRs of the plant *HMGR* genes revealed the presence of conserved regions; these are marked in blue in Figure 7. These regions were detected in both the 5′UTRs and proximal promoters. In most of the tested sequences, PRS was identified within the preserved areas. Additional analysis with the DiAlign TF tool revealed the presence of conserved TFBSs. However, no TFBS was found to be conserved in any of the analysed sequences. The most conserved site was the TATA box, detected in 41.7% of the sequences. Two preserved binding sites for TFs belonging to the P$AHBP (*Arabidopsis* homeobox protein) and P$GCCF (GCC box family) families were detected in 27.8% of cases. Sites for P$TDTF (transposase-derived proteins), P$MYBL (MYB-like proteins) and P$L1BX (L1 box) proteins were identified in 25% of the sequences. Binding sites for P$DREB (dehydration responsive element binding factors) and P$ROOT (root hair-specific *cis*-elements in angiosperms) families were found in 22.2%. These TFBSs are highlighted in red in Figure 7.

Apart from the conserved TATA box and PRS motifs, the investigated *S. miltiorrhiza HMGR4* promoter was found to contain several other common binding sites for TFs, belonging to P$AHBP (*Arabidopsis* homeobox protein), P$GTBX (GT box elements), P$DOFF (DNA binding with one finger) and P$L1BX (L1 box); these were shared by 8–13% of the tested sequences. Nucleotide pairwise alignment of *S. miltiorrhiza HMGR4* with the remaining tested sequences found 13 to 31% identity (mean 24.7%).

## 3. Discussion

Our study presents new data regarding the isolated *S. miltiorrhiza HMGR4* promoter and 5′UTR and compares these sequences with other plant *HMGR*s.

Initially, the sequences were examined for the presence of certain distinctive motifs (Figure 1). One such motif found in the investigated sequences is the TATA box, which is estimated to be present in 30–50% of all known promoters [41] and 29% of *A. thaliana* promoters [42]. It was also detected in the *S. miltiorrhiza HMGR2* promoter [43]. Previous studies in human and yeast models indicate that the TATA box is more common in promoters of highly-regulated genes and in those stimulated by stress factors and extracellular signals [44,45,46,47,48]; in contrast, TATA-less genes are more constitutively expressed and associated with key processes such as cell growth [44,45,46,47,48]. In addition, promoters containing the TATA box appear to have a more conserved sequence than those that do not [49].

The *HMGR4* promoter is also characterised by the presence of a single tandem repeat. This motif is estimated to be present in only 25% of promoters [50], and is absent from the *S. miltiorrhiza HMGR2* promoter [43]. As tandem repeats are more prone to mutation, which affects the length of the repeat and thus local nucleosome positioning and gene expression rate, genes whose promoters have tandem repeats show higher rates of transcription divergence [50].

Both the *HMGR2* promoter and the studied *HMGR4* promoter lack CpG islands [43]. The cytosines in the CG dinucleotides of the islands can be methylated, thus inhibiting gene expression [51,52]. However, the CpG cluster is not required for methylation since, in plants, it can also occur within the CHG and CHH sequences (H = A, T or C) [53].

A PRS is also detected in the 5′UTR of the described sequence. This is a rather rare observation, but not an unprecedented one, as a PRS has also been found in the 5′UTR of the *S. miltiorrhiza HMGR2* gene [43]. It is believed to take part in the organisation of the spliceosomal complex [54].

Furthermore, the examined *S. miltiorrhiza HMGR4* promoter sequence turned out to be rich in TFBSs recognised by specific TFs (Appendix A). The conducted research indicates that the number of promoter regulatory elements and interacting proteins positively correlates with divergence of gene expression [55]. The *HMGR4* proximal promoter was found to contain consensus sequences mainly related to the response to light, SA, bacterial infection, auxins, ABA, gibberellin, chitin, cold or, finally, salt stress (Figure 2), suggesting that these factors may participate in gene regulation. One previous paper investigating the influence of external agents on *S. miltiorrhiza HMGR4* found that treatment with 200 µM MeJa had no significant effect on *HMGR4* expression in either leaves or roots [19]. It is important to note that the effect of these factors has been examined on other *S. miltiorrhiza HMGR* genes. Chen et al. found that only 100% red light slightly increased the expression of *HMGR* in hairy root culture, while other light types (e.g., 100% far-red, 100% blue, red:far-red, blue:far-red, red:blue, red:blue:UV) had an inhibitory effect [56]. In contrast, Wang et al. noted that UV-B enhanced the expression of *HMGR1* in roots almost 5-fold compared to an untreated control [57]. Incubation of hairy root culture with 100 µM SA raised *HMGR* transcript level, peaking at three-times higher than baseline after 36 h [58]. In turn, 200 µM SA has been found to have a differential effect on *HMGR2* promoter in leaf material [43]. A decrease in its activity was observed after 12, 24 and 48 h of treatment, while a 2.5- to 3-fold rise compared to the calibrator values was observed after 72 and 96 h. Bacteria appeared to be good activators of *HMGR* expression. The addition of *Pseudomonas brassicacearum* subsp. *neoaurantiaca* and *Pseudomonas thivervalensis* to *S. miltiorrhiza* root culture resulted in 2.1- and 1.5-fold enhancements in HMGR enzyme activity, respectively [59]. In addition, *Streptomyces pactum* Act12 increased *HMGR1* expression by more than a factor of 35 on day 14 relative to the calibrator [60]. Exposure to 2.85 µM IAA and 2.88 µM gibberellic acid improved the activity of the *HMGR2* promoter, resulting in manifold higher expression compared to the calibrator in 96 h [43]. In turn, 200 µM and 10 µM ABA upregulated *HMGR1* and *HMGR2*, respectively [43,61]. Salt stress (50 mM, 100 mM, 200 mM, 300 mM NaCl) enhanced the expression and enzymatic activity of *HMGR* in leaves and roots over 48 h of exposure [62]. It was also found that 200 mM NaCl inhibited the level of *HMGR1* transcript in leaves and roots as compared to the calibrator [63].

As non-coding regions are generally not highly conserved, from an evolutionary perspective, finding such motifs in the promoter or 5′UTR sequences suggest they have functional importance [64].

Within the studied 36 *HMGR* sequences, the most frequently-identified conserved motif was the TATA box (Figure 7). This is a known sequence that has been conserved from *Archaebacteria* to humans [65]. The other TFBSs discussed in the Results section were shared by a much smaller number of tested sequences (27.8% or fewer).

The study also examined the possibility that more complex structures could be created by TFs interacting with the *HMGR4* promoter. TFs participate in the regulation of gene expression as monomers, dimers (homo- and heterodimers) or multimers. Dimers and multimers are often preferred by nature because they allow specific interactions with the promoter sequence and bind with high affinity [66]. One TF monomer can create dimers or multimers with different functions, thus mediating the regulation of various genes, by forming bonds with multiple, but not random, protein partners [67]. Our analyses revealed the presence of closely-related TFBSs for the following TFs in the *S. miltiorrhiza HMGR4* proximal promoter: HD-ZIP (ATML1, PDF2 and HDG1), WRKY (WRKY2, WRKY14, WRKY45, WRKY57 and WRKY69) and Dof (DOF5.4) (Table 1). The HD-ZIP proteins are unique to the plant kingdom. TFs from the family are unable to bind to DNA as monomers [68]. They form homo- and heterodimers via the leucine zipper motif [67]. Meanwhile, ATML1 was able to create a heterodimer with its paralogue PDF2 in studies on *Nicotiana benthamiana* and *A. thaliana* [69,70], and to form homodimers in vitro [69,71]. It has been shown that WRKY TFs can interact with DNA as monomers or create homo- and heterodimers, especially those with a leucine zipper motif [72,73,74], WRKY2 protein was found to form homodimers in *Hordeum vulgare* [75], while WRKY45 created homodimers in vitro by exchanging β4-β5 strands in *Oryza sativa* [72]. The Dof TFs have a multifunctional domain that allows them to bind to DNA and interact with other proteins [76] and to establish homo- and heterodimers.

As miRNA is believed to regulate plant promoter activity at the transcription level, the investigated *HMGR4* sequence was searched for miRNA binding sites and interacting miRNAs [27]. Of the 12 miRNAs potentially binding to the *HMGR4* promoter and 5′UTR sequences (Table 2), non-conserved miR1128 and miR1436 were detected during deep sequencing in *S. miltiorrhiza* [77]. However, their significance in the regulation of gene expression has not yet been investigated at the experimental level.

The results of our present in silico analysis of the *HMGR4* promoter and 5′UTR sequences constitute a strong basis for planning future necessary experiments on *S. miltiorrhiza*.

## 4. Materials and Methods

### 4.1. Plant Material

*S. miltiorrhiza* plants were cultivated from seeds provided by the Garden of Medicinal Plants of the Medical University of Lodz. The plants were grown in pots containing composite soil at 26 ± 2 °C under natural light. Six-month-old plants were used for the experiment.

### 4.2. Isolation of the S. miltiorrhiza HMGR4 Promoter Sequence

Genomic DNA used for isolation of the *HMGR4* promoter was obtained from young, fresh *S. miltiorrhiza* leaves and stems according to the method proposed by Khan et al. [78]. The DNA was analysed using a NanoPhotometer P300 (Implen, Munich, Germany) to determine its quantity and quality based on A_260_/A_280_ and A_260_/A_230_ ratios. The *HMGR4* promoter region was isolated using GenomeWalker Universal Kit (Takara Bio, Kusatsu, Japan) according to the manufacturer’s instructions. A 5′-terminal fragment of the *HMGR4* gene, deposited in GenBank under accession number JN831103.1, was used as a target for designing GSP1 and GSP2 specific primers (Appendix A). The PCR reactions were performed using the Advantage 2 PCR Kit (Takara Bio, Kusatsu, Japan). The amplified DNA fragments were TOPO-TA cloned into a pCRII-TOPO vector (Invitrogen, Carlsbad, CA, USA) according to the manufacturer’s instructions. The inserts were Sanger sequenced (CoreLab, Medical University of Lodz, Lodz, Poland) with specific primers listed in Appendix A. The *HMGR4* promoter sequence was assembled using CodonCode Aligner software version 8.0.2 (CodonCode Corporation, Centerville, MA, USA).

The isolation and sequencing of the *S. miltiorrhiza HMGR4* promoter took approximately one month.

### 4.3. In Silico Analysis of the S. miltiorrhiza HMGR4 Promoter Sequence

The obtained *HMGR4* promoter sequence was characterised in silico using available tools and databases [79]. The promoter, TATA box, TSS and 5′UTR positions were identified with TSSP software (Softberry Inc., Mount Kisco, NY, USA) [80]. Tandem repeats, CpG islands, TFBSs and TFs were detected using PlantPan 2.0 [81]. The promoter sequence was screened for the presence of commonly-known consensus motifs reported in the published literature. Assuming that the functional TFBSs are concentrated mainly in the proximal promoters, special attention was paid to the promoter region lying within 300 bp from the TSS. The miRBase tool was used to search for miRNA binding sites and interacting miRNAs in the obtained promoter and 5′UTR sequences [82].

### 4.4. Microarray and NGS Co-Expression Data Analysis

Protein BLAST (NCBI, Bethesda, MD, USA) and MEGA X version 10.2.6 (Pennsylvania State University, State College, PA, USA) [83] were employed to determine which of the *A. thaliana HMGR* genes is a homologue of the *S. miltiorrhiza HMGR4* gene. Analyses were performed on coding sequences. Expression Angler (BAR, Toronto, ON, Canada) [84] and *Arabidopsis* RNA-seq Database [85] were utilised to find TF genes co-expressed with the selected *A. thaliana HMGR* gene. The Expression Angler tool has access to the expression results for approximately 22,000 *Arabidopsis* genes, while the *Arabidopsis* RNA-seq Database integrates 28,164 publicly available *Arabidopsis* RNA-seq libraries. The following microarray dataset compendiums were used during the study: AtGenExpress Hormone and Chemical, AtGenExpress Abiotic Stress, AtGenExpress Pathogen, AtGenExpress Tissue, and AtGenExpress Plus—Extended Tissue. The Pearson’s correlation coefficient (*r*) ranging from 0.50 to 1.00 (moderate to strong positive correlation) was applied to identify co-regulated genes. Information on the detected TFs was obtained from the UniProt database [86]. The collected results were compared with the in silico data found by PlantPan 2.0. The occurrence of interactions between the received common TFs was determined using Pathway System (Genomatix, Munich, Germany) and the BioGRID database version 4.4.201 [87].

### 4.5. Comparison of S. miltiorrhiza HMGR Promoters

The entire available *S. miltiorrhiza HMGR* promoter sequences, i.e., *HMGR1* (GU367911.1), *HMGR2* (KF297286.1), and *HMGR4* (KT921337.1) were analysed with Common TFs, FrameWorker and DiAlign TF tools from Genomatix, Munich, Germany. Common TFs was used for preliminary analysis of the common TFBSs and interacting TFs located anywhere in the investigated promoters. The search only included sites that were common to all three sequences. The similarity of the matrix to the tested sequences was set to the highest possible value, i.e., 0.05. The FrameWorker tool permitted the common, most complex framework of TFBSs to be extracted from the input promoters. Frameworks are defined as TFBSs that occur in the same order and in a specificed space range in all of the sequences. The DiAlign TF allowed for multiple alignment of the studied *HMGR* promoters, and revealed conserved regions and TFBSs located therein. The analyses using the Genomatix tools were based on matrix library version 11.3 and default search criteria.

### 4.6. Assessment of Conservation of Plant HMGR Promoters

The conservation of the 36 available *HMGR* promoters derived from plants such as *A. thaliana*, *Arabidopsis lyrata*, *Glycine max*, *Gossypium hirsutum*, *Oryza sativa*, *Solanum lycopersicum*, *Zea mays* and *S. miltiorrhiza* was assessed by aligning their sequences. Proximal promoters and 5′UTR sequences (each sequence 500 bases long) were obtained from PlantPan 3.0 [88] and NCBI Nucleotide databases with the participation of the UniProt [86]. Alignments were performed using the MUSCLE algorithm from the MEGA X software, version 10.2.6 [83]. TFBSs located in the conserved regions of the compared sequences were recognised with the DiAlign TF tool (Genomatix, Munich, Germany).

## 5. Conclusions

Regulation of *S. miltiorrhiza HMGR4* gene expression can occur during flowering, embryogenesis, organogenesis and circadian rhythm, and are influenced mainly by factors such as light, SA, bacterial infection and auxins.

The presence of binding sites for TFs that promote the biosynthesis of tanshinones may indicate that the *S. miltiorrhiza HMGR4* gene plays an important role in the production of these metabolites.

A comparison of TFBSs and TFs in the *S. miltiorrhiza HMGR1*, *HMGR2*, and *HMGR4* promoter sequences indicates that these genes can be co-regulated in response to abiotic and biotic factors, and during organogenesis.

The *S. miltiorrhiza HMGR4* promoter is not highly conserved.

Future research on the *S. miltiorrhiza HMGR4* promoter could be developed towards preparing promoter deletion mutants, and studying their transcriptional activity [89]. Moreover, mutagenesis of particular TFBSs could be suitable for experimental verification of their importance in response to biotic or abiotic factors [89]. The TFs or other regulatory proteins could be isolated using a yeast-one hybrid (Y1H) system and the promoter segments as bait [90]. Isolated TFs could be functionally characterised by studying their DNA binding properties, and their potential to increase expression of specific genes [91,92]. These studies could be verified by chromatin immunoprecipitation-sequencing (ChIP-seq) of DNA fragments that are associated with particular proteins [93]. Finally, regulatory networks of TFs and other proteins playing a pivotal role in the response to certain external factors could be built using transcriptomic RNA sequencing, and weighted gene co-expression network analysis (WGCNA) [94,95].

## Figures and Tables

**Figure 1 plants-11-01861-f001:**
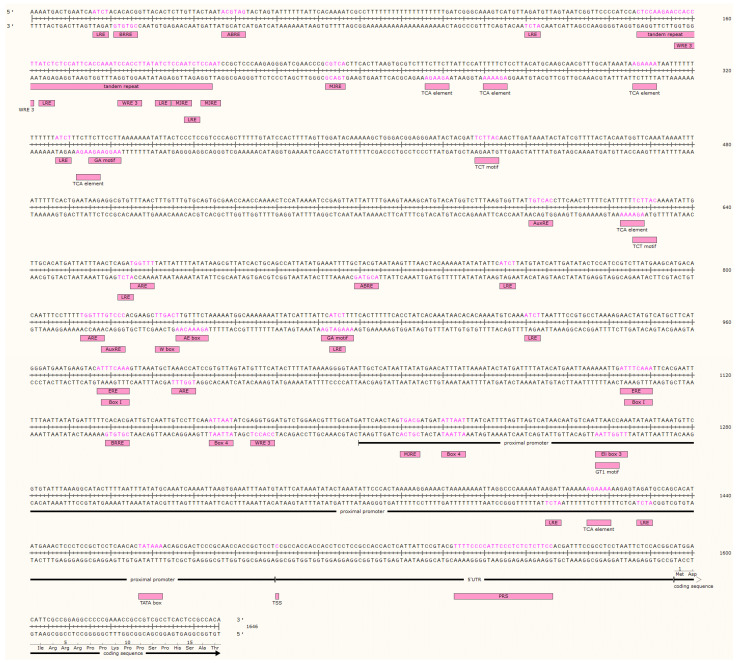
Isolated *S. miltiorrhiza HMGR4* promoter sequence (1499 bp), 5′ untranslated region (5′UTR) (96 bp) and coding sequence fragment (51 bp). The potential TATA box, transcription start site (TSS), pyrimidine-rich sequence (PRS), tandem repeat and consensus sequences for hormone-, pathogen-, wounding-, light-, and anaerobic-responsive elements are signed and marked in pink on the strands. ABRE, abscisic acid-responsive element; ARE, anaerobic-responsive element; AuxRE, auxin-responsive element; BRRE, brassinosteroid-responsive element; ERE, ethylene-responsive element; LRE, light-responsive element; MJRE, methyl jasmonate-responsive element.

**Figure 2 plants-11-01861-f002:**
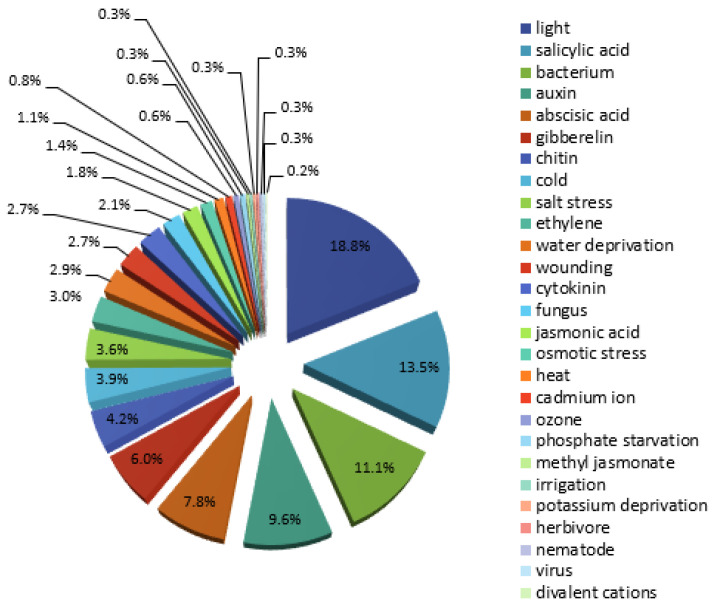
Classification of TFBSs found in the proximal *S. miltiorrhiza HMGR4* promoter with regard to their response to biotic and abiotic factors.

**Figure 3 plants-11-01861-f003:**
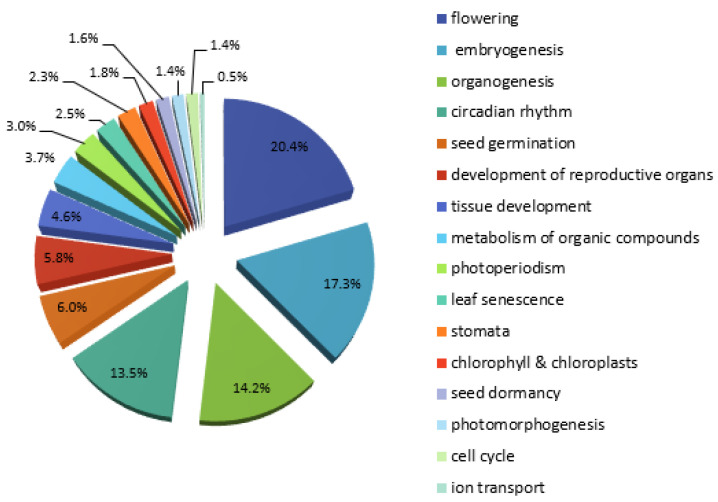
Classification of TFBSs found in the proximal *S. miltiorrhiza HMGR4* promoter with regard to their biological functions.

**Figure 4 plants-11-01861-f004:**
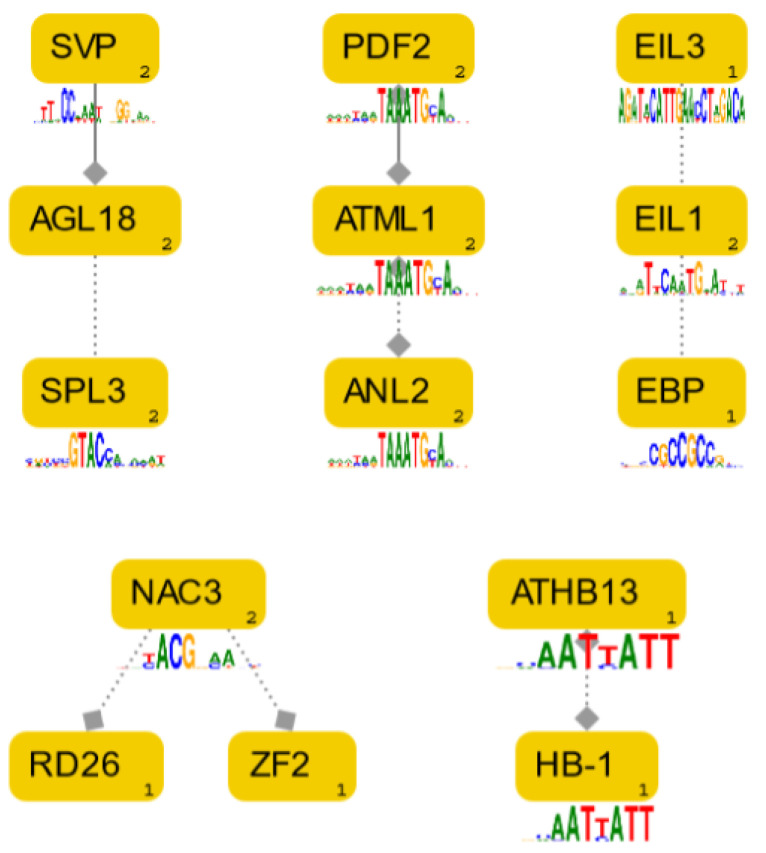
Interactions between TFs potentially binding to the *S. miltiorrhiza HMGR4* promoter found with the Pathway System tool. The presented dependencies are based on co-citation (dashed line) or expert-curation (solid line). Diamond-ended lines indicate that a given TF has a predicted binding site in dependent promoter sequence. The number in the lower right corner of TF indicates the number of interactions within the network (including those not displayed). EBP = RAP 2-3, NAC3 = NAC055, RD26 = NAC072, ZF2 = AZF2, HB-1 = HAT5.

**Figure 5 plants-11-01861-f005:**
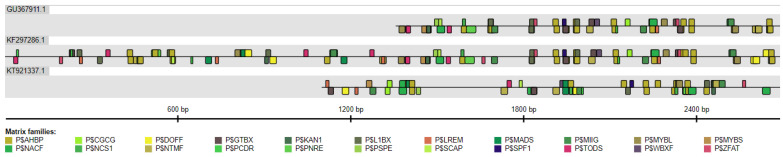
Distribution of matrix families common to the *S. miltiorrhiza* promoter sequences, i.e., *HMGR1* (GU367911.1), *HMGR2* (KF297286.1) and *HMGR4* (KT921337.1) identified by the Common TFs tool. Black lines correspond to the promoter sequences. Each matrix family is marked with a semicircular coloured symbol. The figure shows families found on the positive and negative strands. P$AHBP, *Arabidopsis* homeobox proteins; P$L1BX, L1 box; P$MIIG, MYB IIG-type binding sites; P$DOFF, DNA binding with one finger; P$GTBX, GT box elements; P$MADS, MADS box proteins; P$MYBL, MYB-like proteins; P$MYBS, MYB proteins with single DNA binding repeat; P$NTMF, NAC factors with transmembrane motif; P$NACF, plant specific NAC proteins; P$KAN1, transcription repressor KANADI; P$WBXF, W box family; P$TODS, time-of-day-specific regulatory elements; P$NCS1, nodulin consensus sequence 1; P$SPF1, sweet potato DNA-binding factor with two WRKY domains; P$ZFAT, zinc finger proteins; P$LREM, light-responsive elements; P$PSPE, protein secretory pathway elements; P$CGCG, CGCG box binding proteins; P$PCDR, proteins involved in programmed cell death response; P$PNRE, plant nitrate-responsive elements; P$SCAP, stomatal carpenter.

**Figure 6 plants-11-01861-f006:**
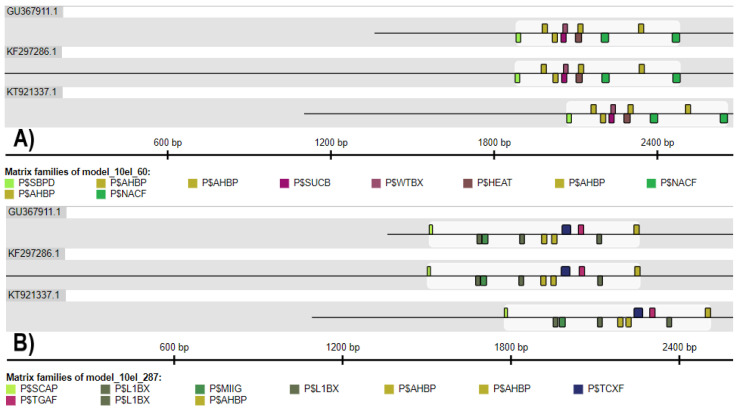
Selected 10-element-frameworks of TFBSs obtained for *S. miltiorrhiza* promoter sequences, i.e., *HMGR1* (GU367911.1), *HMGR2* (KF297286.1) and *HMGR4* (KT921337.1) using the FrameWorker tool. Black lines correspond to the promoter sequences. Each matrix family is marked with a semicircular coloured symbol. The figure shows families found on the positive and negative strands. (**A**) P$SBPD, SBP-domain proteins; P$AHBP, *Arabidopsis* homeobox proteins; P$SUCB, sucrose box; P$WTBX, WT box; P$HEAT, heat shock factors; P$NACF, plant specific NAC proteins; (**B**) P$SCAP, stomatal carpenter; P$L1BX, L1 box; P$MIIG, MYB IIG-type binding sites; P$AHBP, *Arabidopsis* homeobox proteins; P$TCXF, CRC domain containing tesmin/TSO1-like CXC (TCX) factors; P$TGAF, basic/leucine zipper-type TFs of the TGA-family.

**Figure 7 plants-11-01861-f007:**
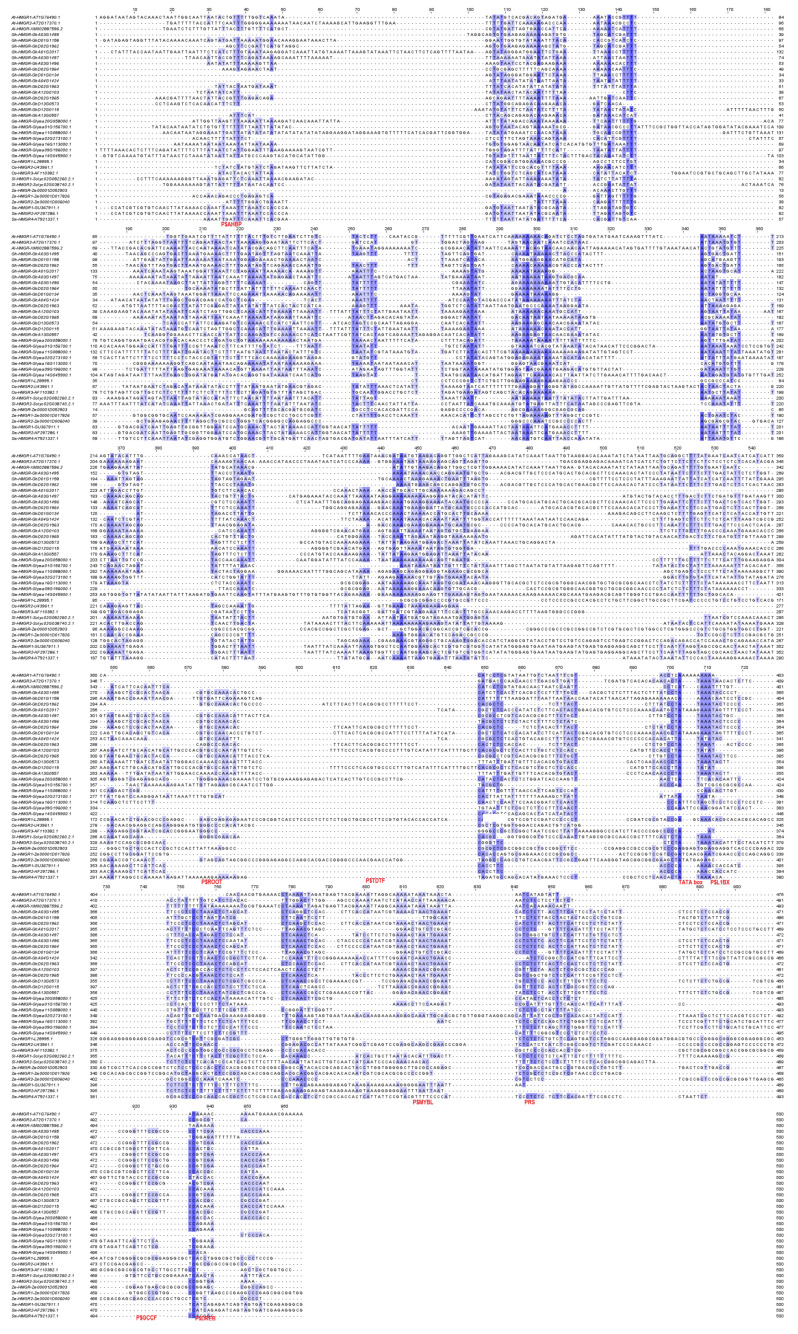
Alignment of proximal promoter regions and 5′UTRs of *Arabidopsis thaliana* (At), *Arabidopsis lyrata* (Al), *Gossypium hirsutum* (Gh), *Glycine max* (Gm), *Oryza sativa* (Os), *Solanum lycopersicum* (Sl), *Zea mays* (Zm) and *Salvia miltiorrhiza* (Sm) *HMGR* genes. Conserved nucleotides are marked in blue. The darker the colour, the greater the degree of conservation within the analysed sequences. The proximal promoters are present at the beginning and the 5′UTRs at the end of the displayed sequences, respectively. Conserved TFBSs identified by DiAlign TF tool are highlighted in red. P$AHBP, *Arabidopsis* homeobox protein; P$GCCF, GCC box family; P$TDTF, transposase-derived proteins; P$MYBL, MYB-like proteins; P$L1BX, L1 box; P$DREB, dehydration responsive element binding factors; P$ROOT, root hair-specific cis-elements in angiosperms.

**Table 1 plants-11-01861-t001:** Transcription factor binding sites (TFBSs) with the potential to form dimers detected in the *S. miltiorrhiza HMGR4* promoter sequence.

Transcription Factor (TF) Family Name	Fragment of Promoter Sequence with Underlined TFBSs ^a^ and Potentially Interacting TF Pairs
˗	AGAAAAATGGAATAAGAAGA two salicylic acid (SA)-responsive TCA elements (AGAAAA and AGAAGA) spaced by eight nucleotides
˗	ATCTCCAATCT two LREs (ATCT) spaced by three nucleotides
HD-ZIP	atTTAATgtaTTCATAAATata ATML1/HDG1 and PDF2/ATML1 spaced by four nucleotides
WRKY	AGTCATAACAATGTCAA WRKY2/WRKY14/WRKY45/WRKY57/WRKY69 and WRKY2/WRKY14/WRKY45/WRKY57/WRKY69 spaced by six nucleotides
Dof	AAAGAAAAAAGA DOF5.4 and DOF5.4 spaced by two nucleotides

^a^ Most conserved positions within a matrix were written in capital letters.

**Table 2 plants-11-01861-t002:** Identification of microRNAs (miRNAs) potentially interacting with the *S. miltiorrhiza HMGR4* promoter sequence and 5′UTR identified by the miRBase tool.

miRNA Name and Source	miRNA Sequence	Sequence Alignment Position Start/End	Strand	e-Value
** *HMGR4* ** **Promoter**				
miR1128 *Saccharum sp.*	UACUACUCCCUCCGUCCCAAA	350/368 405/423	+ −	0.75 4.2
miR6462c-5p *Populus trichocarpa*	AAGGGACAAAAAUGGCAUAAGA	259/279	−	3.5
miR1128 *Triticum aestivum*	UACUACUCCCUCCGUCCGAAA	350/368	+	4.2
miR1436 *Oryza sativa* and *Hordeum vulgare*	ACAUUAUGGGACGGAGGGAGU	354/368	−	6.2
miR5205a *Medicago truncatula*	CAUACAAUUUGGGACGGAGGGAG	355/374	−	9.1
miR8740 *Gossypium raimondii*	UAAUGAUGUGGCACAAUAUUA	634/653	−	9.1
miR11573a and miR11573b *Picea abies*	UUGGGGAGCGUAUUGUAGAUU	197/216	−	9.1
**5′UTR of *HMGR4***				
miR477 *Gossypium raimondii*	CGAAGUCUUGGAAGAGAGUAA	59/75	−	3.2
miR6180 *Hordeum vulgare*	AGGGUGGAAGAAAGAGGGCG	55/69	−	3.9
miR4993 *Glycine max*	GAGCGGCGGCGGUGGAGGAUG	13/30	−	6.9
miR12107-5p *Citrus sinensis*	CUGAUGAGAGAGCGAAUGAUA	51/66	−	8.4

**Table 3 plants-11-01861-t003:** TFs common between in silico analysis of the *S. miltiorrhiza HMGR4* promoter and microarray co-expression studies with *A. thaliana HMGR1*.

TF Family Name	TF Gene Name and Locus	Processes in Which TF is Involved ^a^	r-Value ^b^	TFBS Motif and Localisation ^c,d^	
Homeodomain; HD-ZIP	*ATHB-13;* At1g69780	cotyledon and leaf morphogenesis; primary root development; sucrose-signalling pathway	0.594	ATAAT 310; 309	AATAA 308; 307
	*ATHB-16;* At4g40060	regulation of timing of transition from vegetative to reproductive phase; repression of cell expansion during plant development; response to blue light	0.562	ATAAT 309	
	*HDG1;* At3g61150	maintenance of floral organ identity	0.507	ATTAA 161	**TTAAT 1218; 1331; 1332**
	*ANL2;* At4g00730	regulation of tissue-specific accumulation of anthocyanins; cellular organisation of primary root; cuticle hydrocarbon biosynthetic process; plant-type cell wall modification; root hair cell differentiation	0.524	**TTAAT 1218**	ATTAA 1161
	*ATML1;* At4g21750	cotyledon development; epidermal cell differentiation; seed dormancy and germination	0.583	**TTAAT 1332** **ATTTA** 1057; **1282**	**TAAAT** 987; **1272**; **1346**
	*PDF2;* At4g04890	cotyledon development; epidermal cell differentiation; seed dormancy and germination; maintenance of floral organ identity	0.587	**ATTTA** 1057; **1282**	**TAAAT** 987; **1272**; **1346**
Homeodomain; bZIP; HD-ZIP	*HAT5;* At3g01470	leaf morphogenesis; response to blue light and salt stress	0.521	ATAAT 307; 310	AATAA 308; 307
bZIP	*BZIP25;* At3g54620	positive regulation of seed maturation	0.666	CCACG 822 TACGT 46; 720 ACGTA 47; 721	AACGT 290;1188 ACGTT 291; 1189
WRKY	*WRKY2;* At5g56270	regulation of basal cell division patterns during early embryogenesis; establishment of cell polarity; longitudinal axis specification; pollen development	0.575/ 0.557	TGACT 5; 832 **AGTCA** 111; **1240**	TTGAC 831 **GTCAA** 913; 1147; **1251**
	*WRKY14;* At1g30650	˗	0.576		
	*WRKY57;* At1g69310	response to osmotic stress, salt stress and water deprivation	0.504		
	*WRKY45;* At3g01970	phosphate ion transport	0.515	TTGAC 830; 831 TGACT 5; 832	**AGTCA** 111; **1240** **GTCAA** 913; 1147; **1251**
	*WRKY69;* At3g58710	˗	0.564		
Myb/SANT; ARR-B	*ARR2;* At4g16110	His-to-Asp phosphorelay signal transduction system; expression of nuclear genes for components of mitochondrial complex I; ethylene- and cytokinin-activated signalling pathways; promotion of cytokinin-mediated leaf longevity; root meristem growth; seed growth; stomatal movement	0.552/0.611	AATCT 15; 197; 919 **AGATT 1405**	AATCC 179; 204; 548
	*ARR14;* At2g01760	His-to-Asp phosphorelay signal transduction system; activation of some type-A response regulators in response to cytokinins	0.509/ 0.530		
Myb/SANT; MYB	*MYB6;* At4g09460	response to ethylene, abscisic acid (ABA), indole-3-acetic acid, and *Pseudomonas syringae* pv. *phaseolica*	0.599	ACCTA 886	
MYB-related	*RVE1;* At5g17300	morning-phased TF integrating circadian clock and auxin pathways; regulation of free indole-3-acetic acid level in time-of-day specific manner; negative regulation of freezing tolerance	0.501	ATATC 1166	
	*RVE4;* At5g02840	regulation of circadian rhythm	0.545/ 0.573	ATATC 1166	**GATAT 1215**
EIN3; EIL	*EIL1;* At2g27050	positive regulation of ethylene response pathway; cellular response to iron ion; defence response to bacterium	0.554	TGTAT 374; 759	ATACA 391
	*EIL3;* At1g73730	ethylene response pathway; sulphur metabolic process; cellular response to iron ion	0.525	ATGTA 757	
MADS box; MIKC	*AGL18;* At3g57390	negative regulation of flowering and short-day photoperiodism; pollen development	0.567	TTTCC 804; 801 TTTTG 805 **CAAAA 1396**	**AGAAA** 293; **1402** **GGAAA 1364**; **1362** TTTTT 77; 78; 79, 80; 81; 82; 83; 84
	*SVP;* At2g22540	inhibition of floral transition in autonomous flowering pathway; identity of floral meristem; response to temperature stimulus	0.537	TTTCC 801	**GGAAA 1362**
NAC; NAM	*NAC055;* At3g15500	jasmonic acid-mediated signalling pathway; response to water deprivation	0.594	TACGT 44; 718 CGTA 720 ACAT 644	TTGAC 829 ACGTA 44; 718; 720
	*NAC072;* At4g27410	activator in ABA-mediated dehydration response	0.543	CGTA 720 TTGAC 829	ACAT 644
NF-YB	*NFYB5;* At2g47810	protein heterodimerization activity	0.558	CTAAT 42 ATCGG 102; 131 CCCAT 139 CCAAG 149; 213 CCAAT 195; 202 GTTGG 389 ATGGG 959 ATTGC 1039 CGAAT 1115 **ATTAG 1389** CCTAT 890 TTTGG 811	**TCAAT** 13; 1149; **1253** AATGG 462; 847 **CCACT** 144; 380; **1364** CCATT 169; 266; 701 CCAAC 530 AGTGG 592 ATTGA 766; 1099 **ACAAT** 460; 800; **1247** ATTGT 599; 637; 1145; 1152 **CCAAA** 176; 534; **1261**; **1396** **CAAAT** 177; 469; 897; 1107; 1161; **1262**; **1313**
NF-YC	*NFYC10;* At5g38140		0.523		
TBP	*TBP2;* At1g55520	required for basal transcription (facilitating the recruitment of TFIID to the promoter, forming a preinitiation complex with RNA polymerase)	0.645	ATATA 746; 739 TTTTA 1022; 1078 **TAAAA** 541; **1465**; 1024	**TATAT** 677; **1305**; 743; 736 **ATAAA** 1025; **1466**; **1344**; 298; 439; 1024; **1465** **TTTAT** 669; 1016; 1072; **1297**; **1223**; **1303**; 674; 1077; **1302**
TCP	*TCP21;* At5g08330	positive regulation of circadian clock	0.546	CCCAC 818	
AP2; ERF	*RAP2-3;* At3g16770	cell death; heat acclimation; ethylene-activated signalling pathway; response to cytokinin, jasmonic acid and other organism	0.502	TAAGA 494	
C2H2	*AZF2;* At3g19580	inhibition of plant growth under abiotic stress conditions; negative regulation of ABA signalling during seed germination; positive regulation of leaf senescence; jasmonate early signalling response; response to chitin and water deprivation; plants overexpressing AZF2 have increased sensitivity to salt stress and barely survive under high salt conditions	0.503	**ACACT** 29; **1462**	
Dof	*DOF5.4;* At5g60850	metal ion binding; binding of OBF TFs to OCS elements	0.727	CGTTA 685 AACGT 286; 1184 ACGTT 289; 1187 GCCTT 79 CCTTT 80; 807 AAAGT 109; 982 AAGGG 1031 **AAGGA 1372**	ACCTT 158; 184; 605; 612 GCTTT 369 AAAGC 400; 571 **AAAGA** 939; **1413**; **1420** **AAAGG** 1030; **1289**; **1371** TCTTT 252; 328; 586; 874 TCCTT 275; 338; 806; 1156 **ACTTT** 382; 509; 881; 1021; **1291**
SBP	*SPL3;* At2g33810	promotion of vegetative phase change and flowering; vegetative- to reproductive-phase transition of meristem	0.596	AGTAC 51; 972 GTACA 578; 973 TGTAC 577 GTACT 52 ATACG 43 CGTAG 46	CTTAC 275; 427; 625 CGTCC 360 GGACG 405 CGAAC 524 CTACG 717 CGTAA 720

^a^ The roles of the TFs were assumed based on the UniProt database. ^b^ r-value between the TFBSs found in the *S. miltiorrhiza HMGR4* promoter and those detected in *A. thaliana* ranged from 0.5 to 1.0. ^c^ For TFBSs, only the most conserved positions within a matrix were listed. ^d^ TFBSs localised in proximal promoter region were put in bold.

**Table 4 plants-11-01861-t004:** Common TF matrix families found during in silico analysis of the *S. miltiorrhiza HMGR1*, *HMGR2*, *HMGR4* promoter sequences using the Common TFs tool.

TF Matrix Family	Processes in Which TF Is Involved ^a^
*Arabidopsis* homeobox proteins (P$AHBP)	root, leaf and anther development; seed maturation; meristem initiation and growth; xylem and phloem pattern formation; cell differentiation; determination of bilateral symmetry; transition from vegetative to reproductive phase; glucosinolate metabolic process; response to: auxin, gibberellin, ABA, water deprivation, blue light and salt stress
L1 box (P$L1BX)	cotyledon development; seed germination and dormancy; epidermal cell differentiation; maintenance of floral organ identity
MYB IIG-type binding sites (P$MIIG)	root, seed, stamen and xylem development; cellular cadmium ion homeostasis; gibberellin and flavonol biosynthesis; defence response to fungi; response to: ABA, chitin, salt stress, cold, water deprivation, phosphate starvation, potassium ion and light
DNA binding with one finger (P$DOFF)	secondary shoot, cotyledon and seed development; cell wall modification; cell cycle; gibberellin biosynthesis; response to: SA, auxin, chitin, red light and cold
GT box elements (P$GTBX)	shoot system and stomatal complex development; trichome morphogenesis; seed maturation and germination; cell size and growth; response to: auxin and water deprivation
MADS box proteins (P$MADS)	flower, ovule and seed coat development; seed maturation; meristem structural organisation; transition from vegetative to reproductive phase; short-day photoperiodism; circadian rhythm; response to auxin
MYB-like proteins (P$MYBL)	integument, anther and pollen development; leaf morphogenesis; seed growth and dormancy; endothelial cell proliferation; vacuole organisation; wax biosynthesis; long-day photoperiodism; defence response to bacteria and fungi; response to: SA, brassinosteroid, gibberellin, ABA, jasmonic acid, chitin, salt, water deprivation and cold
MYB proteins with single DNA binding repeat (P$MYBS)	leaf and lateral root development; leaf senescence; circadian rhythm; peroxidase activity; auxin and sulphate ion homeostasis; response to: ABA, phosphate starvation, absence of light and high light intensity
NAC factors with transmembrane motif (P$NTMF)	leaf and trichome morphogenesis; xylem development; seed germination; photoperiodism; membrane protein proteolysis; response to: gibberellin, salt stress
plant specific NAC proteins (P$NACF)	leaf and secondary shoot development; primary shoot apical meristem specification; formation of organ boundary; regulation of timing of organ formation; response to water deprivation
transcription repressor KANADI (P$KAN1)	phenylpropanoid metabolic process
W box family (P$WBXF)	induced systemic resistance; JA-mediated signalling pathway; phosphate ion transport; defence response to: bacteria, fungi and viruses; response to: SA, chitin and wounding
time-of-day-specific regulatory elements (P$TODS)	circadian rhythm; red or far-red light signalling pathway; response to temperature
nodulin consensus sequence 1 (P$NCS1)	nodule-specific expression
zinc finger proteins (P$ZFAT)	regulation of root development; phosphate ion homeostasis
light-responsive elements (P$LREM)	response to hypoxia
protein secretory pathway elements (P$PSPE)	SA induction of secretion-related genes via NPR1
CGCG box binding proteins (P$CGCG)	leaf senescence; defence response to: bacteria, fungi and insects; response to: cold, auxins and water deprivation
proteins involved in programmed cell death response (P$PCDR)	regulation of expression of vacuolar processing enzyme
plant nitrate-responsive elements (P$PNRE)	nitrate assimilation; stomatal movement; response to: nitrate and water deprivation
stomatal carpenter (P$SCAP)	stomatal movement
sweet potato DNA-binding factor with two WRKY domains (P$SPF1)	˗

^a^ The roles of the TFs were assumed based on the MatInspector (Genomatix) database.

## Data Availability

The data presented in this study are available in the article and Appendix A.

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
