# Peer review of "Isolation and Comprehensive in Silico Characterisation of a New 3-Hydroxy-3-Methylglutaryl-Coenzyme A Reductase 4 (HMGR4) Gene Promoter from Salvia miltiorrhiza: Comparative Analyses of Plant HMGR Promoters"

_plants, 2022, doi:10.3390/plants11141861_

Round 1
Reviewer 1 Report
Majewska et al., describe the isolation of new S. miltiorrhiza HMGR4 promoter and 5΄ untranslated region (5΄UTR) sequences, and their in silico characterisation via specialised databases such as PlantPan 2.0, TSSP and miRBase. The obtained results were verified by comparison with available co-expression and literature data. The presence of potential interactions between the detected TFs was evaluated using BioGRID and Pathway System. The TFBSs and TFs in the available S. miltiorrhiza HMGR promoters were compared using Common TFs, FrameWorker and DiAlign TF tools. In addition, the sequences of 36 plant HMGR promoters were aligned with MEGA X and DiAlign TF to determine their conservation. The comprehensive in silico analysis presented here gathers valuable new information about the regulatory functions of HMGR promoters. The data can be used to create modified or synthetic promoters which are active under certain controlled conditions. As a result, larger amounts of medically important metabolites can be obtained. Although the topic is attractive, there are some minor concerns that should be addressed.
Fig. 1 should be replaced with an image of better quality.
The conclusion section is very short. At least it should discuss more future work.
Reviewer 2 Report
In this study, Majewska et al. reported isolation and comprehensive in silico characterisation of a new 3-Hydroxy-3-methylglutaryl-coenzyme a reductase 4 (HMGR4) gene promoter from Salvia miltiorrhiza. Also, the authors make comparative analyses of plant HMGR promoters.
The topic presents interest because may provide a new insight for understanding the mechanisms that direct transcription of the S. miltiorrhiza HMGR4 gene.
It is clear that the authors have done intense research and congratulate them on this. However, it is required additional studies for the paper improvement and possible publication in Plants journal.
My comments are as follows:
- Lines 64-66. Please provide 1-2 more references for this statement.
- As you mentioned from the beginning, the received data were verified by comparison with microarray co-expression results obtained for Arabidopsis thaliana. Therefore, in my opinion, you should briefly describe some of the skills and importance of this plant in the Introduction. This will of course be obvious for people from the field, but maybe these mentions will be helpful for other scientists, from the very beginning, and thus the purpose will be clearer.
- Lines 71-76. I think that these mentions are not necessary in the Introduction part. Please remove them from here and insert them in Material and Method instead.
- I noticed that you refer to Table 2 in subchapter 2.1. But this table appears in subchapter 2.2. Please clarify this.
- Lines 323-325. I think it's appropriate to mention a few references here, because you're referring to ″previous studies″.
- It would be interesting to specify in the Material and Method the period in which the experiments took place.
- Finally, in conclusion, could you say if you see any continuation of that research in the future?
